# Application of Nanoparticles in Bioreactors to Enhance Mass Transfer during Syngas Fermentation

Evelyn Sajeev [1], Sheshank Shekher [2], Chukwuma C. Ogbaga [3,4], Kwaghtaver S. Desongu [5], Burcu Gunes [1] and Jude A. Okolie [6,*]

1   School of Biotechnology and DCU Water Institute, Dublin City University, D09 NA55 Dublin, Ireland
2   Department of Chemistry, Netaji Subhas University of Technology, Delhi 110078, India
3   Department of Biological Sciences, Faculty of Natural and Applied Sciences, Nile University of Nigeria, Abuja 900001, Nigeria
4   Department of Microbiology and Biotechnology, Faculty of Natural and Applied Sciences, Nile University of Nigeria, Abuja 900001, Nigeria
5   Department of Chemical Engineering, Federal University of Technology, Minna 920101, Nigeria
6   Gallogly College of Engineering, University of Oklahoma, Norman, OK 73019, USA
*   Correspondence: jude.okolie@ou.edu

**Definition:** Gas–liquid mass transfer is a major issue during various bioprocesses, particularly in processes such as syngas fermentation (SNF). Since SNF involves the movement of gases into the fermentation broth, there is always a rate-limiting step that reduces process efficiency. Improving this process could lead to increased efficiency, higher production of ethanol, and reduced energy consumption. One way to improve fluid transfer between gas and liquid is by incorporating nanoparticles (NPs) into the liquid phase. This entry describes recent advances in using NPs to improve gas–liquid mass transfer during SNF. The entry also describes the basics of SNF and the impact of NPs on the process and suggests areas for future research. For example, carbon nanotubes have been found to elevate the available surface area needed for gas–liquid transfer, thus improving the process efficiency. Another area is the use of NPs as carriers for enzymes involved in syngas fermentation.

**Keywords:** nanoparticles; syngas fermentation; mass transfer; biofuel; bioethanol

## 1. Introduction

Climate change, rising global population, and the ongoing need for energy have driven research into alternative energy sources in recent years. Technologies that convert biogenic waste into green fuels and chemicals, such as thermochemical processes including pyrolysis and gasification, and biological processes such as anaerobic digestion and syngas fermentation, show promise as viable alternatives [1]. Among biological processes, syngas fermentation (SNF) is a particularly promising technology for the production of ethanol from lignocellulosic biomass. SNF has the advantage of not requiring biomass pretreatment, and it is a viable alternative to Fischer-Tropsch Synthesis (FT) for the production of liquid hydrocarbon fuels. It has been studied extensively and has the potential for industrial-scale applications. Unlike FT, SNF does not require a fixed $CO/H_2$ ratio [2]. SNF can also be combined with thermochemical processes in a hybrid process that involves gasifying the feedstock for syngas production and subsequent microbial action of the produced syngas for bioethanol production [3].

One of the major challenges in implementing SNF on a large scale is the low mass transfer rate at the gas–liquid interface [4]. To overcome this limitation, an efficient bioreactor configuration and other key factors are required to ensure a successful mass transfer. However, even with an optimized bioreactor, the process may still be limited by a low rate of mass transfer that cannot meet the demands of cell growth.

The key bottleneck in SNF is how to move the gas molecules to the fermentation broth which is mostly in liquid form. The mass transfer restrictions between gas and liquid often induce low yield and process heterogeneity [2]. Therefore, a bioreactor configuration that can produce efficient mass transfer and a high cell density in a cost-effective manner is crucial for SNF. Common reactors such as the continuous stirred tank reactor (CSTR), bubble column, and airlift reactors are usually adopted in SNF to overcome mass transfer limitations [5].

The volumetric gas–liquid mass transfer coefficient ($k_{La}$) is commonly utilized in evaluating the mass transfer efficiency among different reactor configurations. While various reactor designs have been explored to improve the performance of SNF, the options for altering reactor design are limited. Alternative methods such as using nanoparticles (NPs) have shown promising potential for enhancing mass transfer in syngas fermentation [4].

Kim et al. [6] conducted a study where they tested six nanoparticles to improve gas–liquid mass transfer during SNF. The nanoparticles tested are made up of carbon-based materials, palladium and alumina-based materials. Their results indicated that silica nanoparticles with 0.3 wt.% showed the best enhancement of SNF. Mass transfer coefficient improvement resulting from the adhesion of NPs to the gas–liquid interface was further clarified based on three distinct mechanisms: shuttling or grazing effect, hydrodynamic effects at the gas–liquid boundary layer, and changes in the specific gas–liquid interfacial area. Additionally, an easy and affordable recovery method is essential for making the process economically viable. Magnetic nanoparticles (MNPs) are a promising option for easy recovery of the nanoparticles [6].

In another study, Kim et al. [7] evaluated the influence of MNPs on CO, $H_2$ and $CO_2$ solubility as well as the acid and alcohol production during SNF [7]. Based on their observations, the magnetic silica nanoparticles with Co and Fe oxides improved the gaseous solubility and production of alcohols and acids compared to the experiments without MNPs.

Given the impact of MNPs on SNF, it is crucial to comprehend the underlying mechanism. However, research in this field is limited. Sun et al. [8] provided a comprehensive review of SNF with a focus on process development but the authors did not discuss the role of MNPs in detail [8]. Recently, Gunes [2] outlined the current status and prospects of biofilm reactors for enhancing higher syngas fermentation yields. Although MNPs were discussed briefly, more information is still lacking in the literature. To fill the knowledge gaps, the present review outlines the advances and progress in MNPs applications for the improvement of gas-liquid mass transfer limitations during SNF. A brief overview of SNF is outlined as well as the effects of MNPs on the syngas fermentation process. It should be mentioned that information about the type of nanoparticles, shapes and detailed information about the process of producing various magnetic NPs as well as their respective composites are not within the scope of this review. Such information has been meticulously described elsewhere [9].

## 2. Overview of Syngas Fermentation

SNF is a biological process employed to convert syngas into environmentally friendly fuels and chemicals by using microorganisms in an oxygen-deficient environment [10]. Syngas represents a combustible mixture of several distinct gases comprising carbon dioxide ($CO_2$), carbon monoxide (CO) and hydrogen ($H_2$). Unlike other processes, it does not require high temperatures or pressures. Additionally, a variety of microorganisms can participate in syngas fermentation by utilizing CO and/or $CO_2/H_2$ as their metabolic building blocks. The primary bacteria used in syngas fermentation are known as "acetogens" and use the acetyl-CoA pathway to convert syngas major gases into acetyl-CoA [11]. SNF can involve a variety of microorganisms, specifically a group of prokaryotic single-cell organisms called "acetogens." These microorganisms utilize the Wood–Ljungdahl pathway, also referred to as the acetyl-CoA pathway, to convert CO, $CO_2$ and $H_2$ into acetic acid. Acetyl-CoA is an intermediate metabolite in this pathway, which can be used to synthesize

cell mass, organic acids and alcohols such as acetic acid, ethanol and complex chemicals. The acetic acid can be released from the cell or reduced to ethanol via acetaldehyde, as shown in Figure 1 [12].

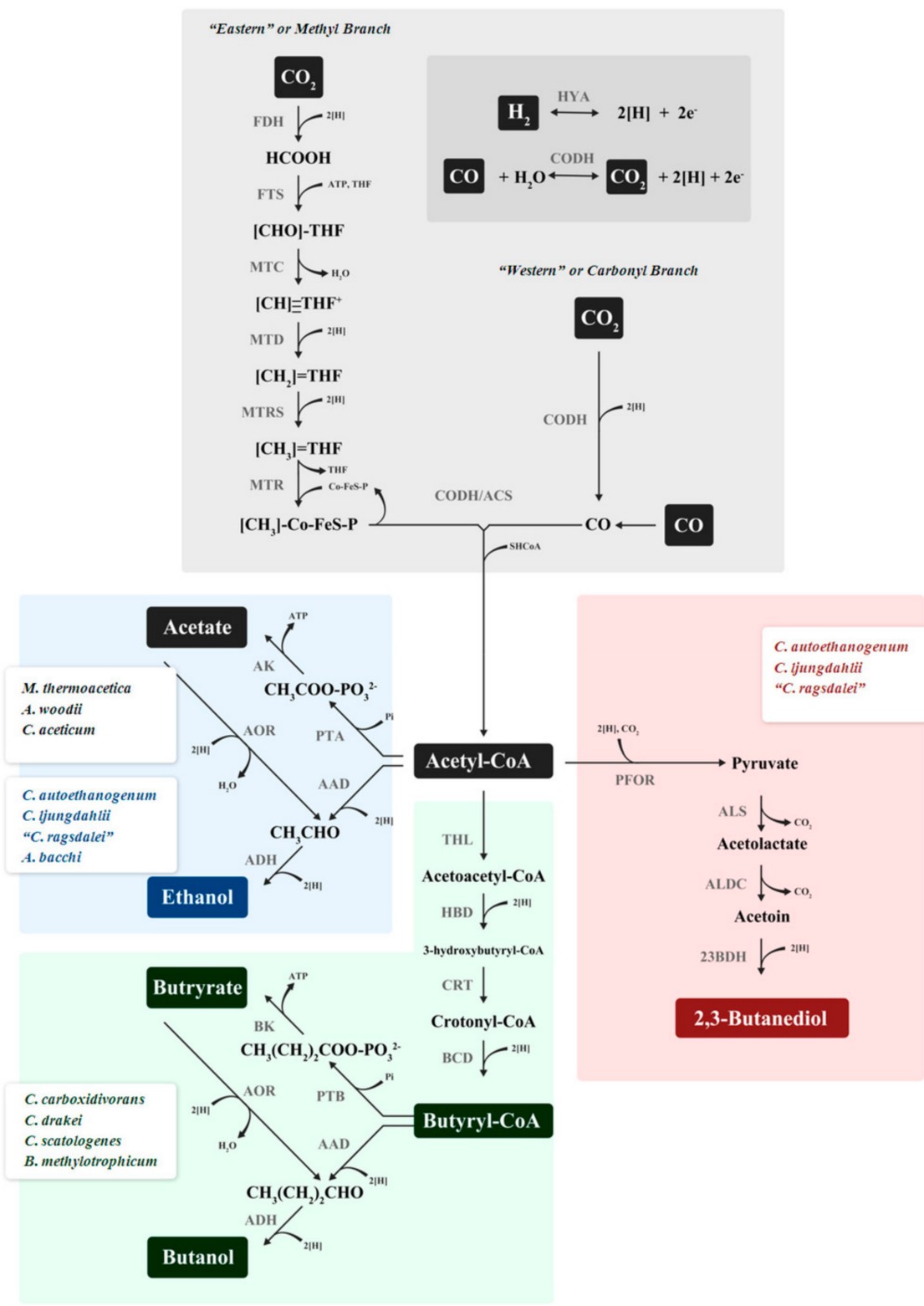

**Figure 1.** Wood–Ljungdahl pathway of acetogens and their metabolic end products. Adapted with permission from Reference [10]. Copyright 2022 Elsevier. Figure colors were used to distinguish between different pathways for the formation of various products.

*Clostridium ljungdahlii, C. autoethanogenum* and *C. carboxidivorans* (also known as aceto-gens) are the few pioneering microbes used for the transformation of syngas into ethanol. They also participate in alternative pathways for the production of butanol and hexanol [11]. To make SNF economically feasible and scalable, it is important to have microbial biocatalysts with specific traits such as high substrate utilization, high product yield, high product selectivity, low product inhibition, prolonged metabolic viability and safety [7].

## 2.1. Effect of Process Parameters on Syngas Fermentation

SNF temperature plays a crucial role in cell metabolism and growth. Temperature influences the syngas key gaseous component solubility when present in the bulk of the liquid. The optimum temperature for microorganisms varies in syngas fermentation, as seen in the study by Shen. [5]. Similarly, the media's pH also influences the metabolic processes, cell growth and product distribution to a very large extent [5]. It has been demonstrated that a shift in the pH could be a promising strategy towards improving the production of ethanol [13].

Richter et al. [14] conducted a two-stage fermentation using *C. ljungdahlii* with two reactors for better ethanol production. The system comprised a 1L CSTR for growth and a 4 L bubble column for ethanol production. Reactor A had a pH of 5.0 (1 L volume) and Reactor B had 4.0–4.5 (1 L volume). This pH shift resulted in a thirty-fold improvement in ethanol productivity compared to a single CSTR which uses *C. ljungdahlii* [14]. This is because the operating pH as well as the temperature can be fixed individually in each stage. Furthermore, the working volume of Reactor A and Reactor B can be changed to various growth and dilution rates to promote rapid growth and acidogenesis [14].

Syngas composition is also vital during SNF. Many microorganisms can use CO as the only carbon and energy source. However, it is believed that syngas fermentation with the presence of $H_2$ can be useful for biofuel production. This is because electrons and protons required for the acetyl-CoA pathway could be acquired from $H_2$ oxidation through hydrogenase or the oxidation of CO by the CODH enzyme. Excess $H_2$ has been shown to enhance ethanol production in *C. ljungdahlii* culture [15]. Syngas produced from biomass conversion processes could also contain several impurities such as sulphur gas, ethane, tar, ethylene and char [15]. These impurities negatively impact syngas fermentation through cell dormancy, inhibition of hydrogenase and cell growth. Consequently, chemical absorbing units such as sodium hydroxide, sodium hydrochloride and potassium permanganate can mitigate the negative impacts of syngas impurities.

## 2.2. Mass Transfer Issues during Syngas Fermentation

The current difficulties of mass transfer in biological systems are the low solubility of synthesis gas components in gas–liquid mass transfer. This is because each biological system is different. Microorganisms have an array of physiochemical and biological differences. Some are filamentous while some can grow branched or dispersed. Some microbes can also increase density and viscosity with time. The mass transfer usually takes place in more than one phase [5].

The volumetric mass transfer ($k_{La}$) is a parameter that characterizes the mass transfer properties in bioreactors. $k_{La}$ is dependent on the bioreactor type and geometry, liquid and gas velocities and fluid properties. In a bioreactor, gas hold-up determines the gas dwell time in the liquid and affects bubble size, which influences the gas–liquid interface for mass transfer. It also impacts bioreactor design, as the maximum gas hold-up dictates the design volume [12]. Furthermore, increased gas hold-up enhances the mass transfer area. A higher superficial gas velocity in the riser leads to a faster liquid velocity, reducing the gas–liquid boundary layer's thickness and decreasing mass transfer resistance [12,16].

The current limitation in mass transfer means it is not high enough to meet the rate of cell growth. Mass transfer limitation makes the availability of substrate too low to be consumed by microbes, resulting in low productivity. Two traditional techniques for

boosting $k_{La}$ values in syngas components include elevating the syngas flow rate and expanding the gas–liquid contact area through faster mixing speeds [17].

In a study conducted in CSTR, a common bioreactor and syngas fermentation were used. Two microorganisms were suspended in the fermentation broths as biocatalysts. The authors observed that raising the specific CO flow rate from 0.14 to 0.86 and enhancing the agitation speed from 200 to 600 rpm increased the $k_{La}$ in the CSTR from 10.8 h$^{-1}$ to 15.5 h$^{-1}$ [17].

It should be noted that escalating the agitation for commercial-scale reactors is deemed economically impractical due to excessive energy expenses. Moreover, raising the syngas flow rate results in the squandering of the gaseous substrate and induces shear stress on microorganisms. Moreover, at a high range of flow rate, the syngas supply could exceed the cells' maximum capability of syngas utilization. Subsequently, the flow rate is also dependent on the type of bioreactor used [4].

## 3. Nanoparticles Classification of Synthesis Method

Nanotechnology has diverse applications and is defined as science at the nanoscale. NPs are particles that are 1–100 nanometers in diameter and are predominantly applied in areas such as energy and biomedicine. NPs are "the building blocks" of nanotechnology [18,19]. Interestingly, NPs are now studied to enhance mass transfer in microbiological processes such as syngas fermentation. This is regarded as a promising strategy to increase mass transfer rates as it provides a large surface area for bacteria and holds the potential to increase the interactions between the liquid and the gas phase [18].

There are different methods of preparing NPs which are broadly classified into top-down methods and bottom-up methods [19]. These methods, which are primarily distinguished by their starting material, tend to tremendously influence the morphology (shape and size) of the nanomaterials formed, as well as their functionalities. In top-down methods, particles of bulk materials are broken into nanoparticles of desired properties and morphology using synthesis techniques such as chemical etching, mechanical milling, sputtering, laser ablation and electro-explosion [20,21]. However, in the bottom-up methods, nanoparticles are synthesized from smaller particles such as atoms and molecules, which act as building blocks [19]. Bottom-up methods include supercritical fluid synthesis, spinning, sol-gel process, laser pyrolysis, chemical vapor deposition, molecular condensation, chemical reduction and green synthesis [20].

Specifically, magnetic nanoparticles (MNPs) have unique properties that make them fit for various applications in areas such as catalysis, biomedicine, magnetic fluids, data storage, environmental remediation, spintronics and magneto-resistance sensors [20]. The properties of MNPs include a high surface-area-to-volume ratio, quantum properties and the ability to carry other compounds, such as drugs, due to their small size. Magnetic fields, whose effectiveness depends on the particle magnetic moment and the field gradient, can be used to manipulate the properties of MNPs to make them suitable for many applications [22,23]. Compared to other nanoparticles, magnetic nanoparticles have several advantages for use as catalyst support in syngas fermentation. The ease of separation, compatibility with different microbial culture and cost-effectiveness are some of the advantages of MNPs when used to enhance SNF [22,23]. The mechanism of MNPs during syngas fermentation involves a series of steps promoted by the improved surface area, magnetic ability and surface chemistry of the nanoparticles. MNPs are characterized by a high surface area, which provides a medium for microbial cells to attach and grow, resulting in more active sites and improved mass transfer [24]. Furthermore, the magnetic properties of the nanoparticles ensure that they are influenced by an external magnetic field, which can increase mixing and improve mass transfer within the fermentation mixture. Generally, the mechanism responsible for MNP-enhanced SNF is driven by their unique properties and excellent surface chemistry.

The best-performing magnetic nanoparticles, depending on the material, have sizes around 10–20 nm because the particle becomes a single domain and exhibits superparamagnetic behavior beyond a temperature called the blocking temperature [23]. This, however, also results in intrinsic instability over longer periods and loss of magnetism that is caused by the oxidation of naked metallic nanoparticles, which are chemically highly active [22,23]. Spherical and cubic magnetic nanoparticles, in particular, have unique desirable properties that have made them objects of much interest [2]. MNPs can be classified into transition or rare-earth metals, alloys and oxides. Transitions metals include Fe, Ni, Co, Gd and so on; alloys include Fe-Co, Fe-Ni, Fe-Ni-Mn, Fe-Pt, and so on; oxides include $Fe_3O_4$, $Fe_2CoO_4$, $Fe_2Mn_xZn_{1-x}Fe_4$, etc. [25]. The most common and useful magnetic materials are based on metal oxides such as iron (Fe), cobalt (Co) and nickel (Ni). However, these have not been fully studied because they have very active surfaces at the nanoscale [25,26]. At the moment, the most utilized magnetic NPs by several researchers is iron oxide [27]. Fe-based magnetic NPs find useful applications in energy and environmental applications [28].

## 4. Review of Nanoparticles for Enhancing Syngas Fermentation

Magnetic NPs play critical roles towards enhanced SNF and will be discussed in this section [29]. Kim and co-workers [6] applied six different types of nanomaterials to improve the mass transfer during syngas fermentation and discovered that the mass transfer of CO, $CO_2$ and $H_2$ were boosted by 272.9%, 200.2% and 156.1%, respectively. The authors confirmed from their study that enhancement of mass transfer through the application of nanoparticles could improve the productivity of fermentation using syngas substrates. In another study, some researchers applied methyl-functionalized silica and methyl-functionalized cobalt ferrite-silica ($CoFe_2O_4@SiO_2$-$CH_3$) nanoparticles to enhance the mass transfer between syngas and water, with the latter showing better improvement. The authors discovered from the study that both nanoparticles did not only significantly improve the rate of mass transfer between syngas and water, but they also maintained their capability to enhance mass transfer after being reused up to five times [7]. In addition, nanoparticles such as spherical MCM41 and functionalized silica nanoparticles have demonstrated the ability to improve the volumetric mass transfer coefficient [30,31].

Applications of NPs in SNF also have the ability to impact the distribution and composition of final products. Prior research study [32] has documented that MCM41 NPs could enhance $H_2$ concentration in the final SNF product. The authors reported the use of microbes called *Rhodospirillum rubrum* [32]. An overview of previous studies that have implemented NPs for improving SNF is summarized in Table 1.

**Table 1.** An overview of previous studies on the application of nanomaterials for syngas fermentation.

| References | Key Findings |
| --- | --- |
| Kim and Lee [7] | The authors compared the performance of two types of nanomaterials including functionalized silica and cobalt ferrite-silica towards enhancing SNF. They used *Clostridium ljungdahlii* microorganism and bioethanol was the target product. The latter showed a better ability to enhance syngas-water mass transfer and more efficient productivity. The nanomaterials retained their ability to enhance mass transfer even after being retrieved and reused for up to five cycles. |
| Kim et al. [6] | Six types of nanomaterials were tested for the production of bioethanol via SNF. Silica nanoparticles at 0.3 wt.% offered better enhancement of mass transfer and increased the level of bioethanol and acetic acid production. |

**Table 1.** *Cont.*

| References | Key Findings |
|---|---|
| Zhu et al. [32] | The authors added the MCM41 nanoparticles with or without mercaptopropyl functional groups to syngas fermentation reactors. This facilitated the fermentation of CO using *Rhodospirillum rubrum* and enhanced the concentration of $H_2$ in the product gas. The yield of $H_2$ was enhanced by about 200% at 0.6 wt.% of the MCM41 nanoparticles |
| Zhu et al. [31] | Spherical MCM41 nanoparticles were designed to enhance volumetric mass transfer coefficient ($k_{La}$) for the fermentation of syngas. These nanoparticles showed a higher value of $k_{La}$ than silica particles, with surface hydroxyl groups playing a vital role in the $k_{La}$ enhancement. Mercaptan groups grafted to MCM41 enhanced the $k_{La}$ by about 1.9 times more than when nanoparticles are not used. |
| Jeon et al. [30] | The authors synthesized silica and methyl-functionalized silica nanoparticles which enhanced the $CO_2$/water mass transfer system. The volumetric mass transfer coefficient experienced a respective rise of 31% and 145%. for each of the nanomaterials, resulting in increased production of bioethanol from fermentation using *Chlorella vulgaris.* |
| Jack et al. [33] | Effluent from $CO_2$ electrolyzer was connected to a bioreactor where the blend of $CO_2$ and CO was converted to acetate and ethanol by *Clostridium ljungdahlii* at rates of $17.87 \pm 7.1$ and $3.23 \pm 1.4$ mg/L/h, respectively, under autotrophic conditions. These production rates were respectively increased by 217% and 224% by the addition of mercapto-modified silica nanoparticles. |
| Gupta and Chundawat [34] | Biologically synthesized ZnO nanoparticles were used to catalyze bioethanol production by the fermentation of sugar obtained from rice straw. A maximum ethanol yield of 0.0359 g/g of dry weight-based plant biomass was produced at a 200 mg/L concentration of ZnO nanoparticle. |
| Sanusi et al. [23] | The researchers investigated the impact of incorporating NPs at various stages during the simultaneous saccharification and fermentation of waste potato peels. Adding NiO nanobiocatalysts at the pre-treatment phase led to a 1.60-fold increase in bioethanol concentration and a 2.10-fold reduction in acetic acid concentration. |

## 5. Conclusions

Syngas fermentation is a promising biological process for the production of biofuels because it does not require biomass pretreatment. In addition, it is a feasible alternative to Fischer–Tropsch synthesis (FT) for the production of liquid hydrocarbon fuels. Syngas fermentation is a major topic among both experimental and modelling researchers. In addition, it has been studied on a commercial scale over the past few years. Compared to FT, syngas fermentation can proceed effectively without a fixed $CO/H_2$ ratio. Several studies have been carried out to circumvent the issues of syngas fermentation, including poor mass transfer issues, low gas solubility and low productivity. The gas–liquid mass transfer of gaseous substrates (CO, $CO_2$ and $H_2$) into the fermentation broth is a rate-limiting step in SNF that leads to low productivity and poor economic feasibility. The addition of magnetic nanoparticles (MNPs) in the liquid phase helps to address the gas–liquid mass transfer limitations thereby achieving an enhanced gas–liquid mass transfer. This mini-review summarized advances in the application of MNPs for improving syngas fermentation. An overview of syngas fermentation process as well as the effect of different operating parameters were briefly discussed. Previous studies in MNPs enhanced syngas fermentation were also reviewed.

Nanoparticles, such as carbon nanotubes, have been shown to increase the surface area available for gas–liquid mass transfer, and thus improve the efficiency of the process. Another area of research is the use of nanoparticles as a carrier for enzymes involved in the syngas fermentation process. Enzymes such as hydrogenases and carbon monoxide dehydrogenases are essential for the conversion of syngas to biofuels and chemicals. By incorporating these enzymes onto nanoparticles, researchers aim to improve their stability and activity and thus enhance the efficiency of the process. Additionally, the use of nanoparticles to immobilize the microorganisms in the bioreactor has also been studied. By immobilizing the microorganisms, they can be prevented from flowing out of the bioreactor and they can be reused, reducing the costs and increasing the efficiency of the process. Overall, research in the application of nanoparticles in bioreactors for syngas fermentation is still in the early stages, but it has shown promising results and has the potential to improve the efficiency and cost-effectiveness of the process. Future studies should also focus on a detailed understanding of the mechanism involved in MNP-enhanced syngas fermentation.

**Author Contributions:** Conceptualization, E.S., S.S., K.S.D. and J.A.O.; methodology, E.S., S.S., C.C.O., K.S.D., B.G. and J.A.O.; investigation, E.S., S.S., C.C.O., K.S.D., B.G. and J.A.O.; software and data processing, J.A.O.; writing original draft, E.S., S.S., C.C.O., K.S.D., B.G. and J.A.O.; writing review draft, C.C.O. and J.A.O. project administration, J.A.O. All authors have read and agreed to the published version of the manuscript.

**Funding:** This research received no external funding.

**Data Availability Statement:** All data are contained within the entry.

**Conflicts of Interest:** The authors declare no conflict of interest.

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
