# Peer review of "Application of Nanoparticles in Bioreactors to Enhance Mass Transfer during Syngas Fermentation"

_encyclopedia, doi:10.3390/encyclopedia3020025_

Round 1
Reviewer 1 Report
Comments and Suggestions for Authors
Manuscript ID: encyclopedia-2213730
Subject: “Application of nanoparticles in bioreactors to enhance mass transfer during syngas fermentation”
This manuscript was shown that the application of nanoparticles to enhance mass transfer. The limitation of syngas fermentation was given, and one of the most suitable methods, that is, nanoparticles to improve the reaction surface area, was chosen. And then, MNPs were also involved. The application of NPs was concluded. Some questions should be considered as follows.
Question:
[1] Page 1, Lines 41-44
The first sentence told us that the microorganisms were suitable for SNF. The second sentence was told that if mass transfer rate at the gas-liquid interface was increased and SNF can be improved. What is relationship between microorganisms and mass transfer rate at the gas-liquid interface? The linked sentences were lacked between the first and second sentence.
[2] Page 3, Table 1
The name “T” with subscript was not completed.
[3] Page 5, Lines 145-146
Are the volumetric mass transfer k with subscript “La” (Line 145), “KLa” (Line 146), and “(KLa)” (Line 54) the same?
[4] Page 6, Lines 191-213
There were two methods to prepare NPs. And MNPs had many special characteristics. Why were MNPs chosen? And what was the relationship between the two paragraphs (Lines 180-213)? Moreover, the application and mechanism of MNPs on mass transfer should be shown.
[5] Page 6-7, “4. Application of nanoparticles during syngas fermentation”
This part showed the application of nanoparticles on mass transfer. What’s important was that the mechanism should be conclude and some suggestions can be shown.
Some other mistakes should be revised:
[1] Page 3, Line 107
Superfluous period “.” behind the word “Safe” should be deleted.
Reviewer 2 Report
Comments and Suggestions for Authors
2 February 2023
Manuscript ID: encyclopedia-2213730
Title: Application of Nanoparticles in Bioreactors to Enhance Mass Transfer
during Syngas Fermentation
Author: Evelyn Sajeev, Sheshank Shekher, Chukwuma Ogbaga, Desongu
Kwaghtaver, Burcu Gunes, Jude A Okolie
In their work, the authors review recent advances in using nanoparticles (NPs) to improve gas-liquid mass transfer during syngas fermentation (SNF).
The report also describes the basics of SNF and the impact of NPs on the process and suggests areas for future research.
Below I present some editorial errors:
1. In lines 32,50, 53, 101 and 107 delete dots before the references.
2. In line 54 and 146 – write subscript kLa
3. In line 64 – add the space after SNF and explain “wt%”
4. In line 102 change “was “ on “were”
5. Table 1 – in the second and three rows, the value of temperature and pH must be changed.
6. Line 119, 123 – add the space after “C. “
7. Line 121 Reactor A and Reactor B write which volumes they have (1L) (4L)?
8. Line 151 – the pause in “hold-up”
9. Line 207 write subscript
10. Line 212 – to refernece [24] or [25] you can add
Miłek J., Tatarchuk T., Modified magnetite nanoparticles synthesized using cetyltrimethylammonium bromide and their application to immobilize trypsin. Biocatalysis and Agricultural Biotechnology 47 (2023) 102586
11. Line 235 write “application” with a small letter
12. Table 2 in the second, four, sixth and seven raws write in italics the name microorganism
13. Table 2 in four and five raws write subscript “H2” and “kLa”
14. Table 2 in six and seven raws write subscript “CO2”
15. Line 241 “Delete n in word “Inn”
16. Line 246 subscript “CO2, and H2)”
17. In References – the name of the microorganism write italics (lines: 280,298,326).
18. In References – chemical compounds written with subscripts (lines: 322, 332)
19. In Reference [4] consider writing the title to lower case.
Conclusion: I recommend this manuscript to publish in Encyclopedia.
Reviewer 3 Report
Comments and Suggestions for Authors
The theme of the review article is very interesting and the authors have written the paper nicely.
A very few typo errors (like operation) have been observed. The authors can go through the paper and correct them once
It would very informative if the authors include some more information about types of nanoparticles, shapes etc.
What is the reason behind enhancing mass transfer due to addition of nanoparticles? What is the mechanism involved for the increase of mass transfer?
Round 2
Reviewer 1 Report
Comments and Suggestions for Authors
The questions have been revised. And the revised manuscript can be considered with publication.
Reviewer 2 Report
Comments and Suggestions for Authors
17 March 2023
Manuscript ID: encyclopedia-2213730
Title: Application of Nanoparticles in Bioreactors to Enhance Mass Transfer
during Syngas Fermentation Author: Evelyn Sajeev, Sheshank Shekher, Chukwuma Ogbaga, Desongu
Kwaghtaver, Burcu Gunes, Jude A Okolie
Thank you for anwers for comments. I have only two comments for manuscript-v.2
1. In line 54 and 147 – write subscript kLa
2. Complete the page 3 to full text or the Figure 1 move on this page.
Conclusion: I recommend this manuscript to publish in Encyclopedia.
Reviewer 3 Report
Comments and Suggestions for Authors
The article could be accepted in the present form